# Microstructure, Physical and Biological Properties, and BSA Binding Investigation of Electrospun Nanofibers Made of Poly(AA-co-ACMO) Copolymer and Polyurethane

**DOI:** 10.3390/molecules28093951

**Published:** 2023-05-08

**Authors:** Hanaa Mansour, Samia M. Elsigeny, Fawzia I. Elshami, Mohamed Auf, Shaban Y. Shaban, Rudi van Eldik

**Affiliations:** 1Chemistry Department, Faculty of Science, Kafrelsheikh University, Kafrelsheikh 33516, Egypt; mansourhanaa@yahoo.com (H.M.); samia.elsigeny@sci.kfs.edu.eg (S.M.E.); fawzyaalshamy@yahoo.com (F.I.E.); mohamed.auf@sci.kfs.edu.eg (M.A.); 2Department of Chemistry and Pharmacy, University of Erlangen-Nuremberg, 91058 Erlangen, Germany; 3Faculty of Chemistry, Nicolaus Copernicus University in Torun, 87-100 Torun, Poland

**Keywords:** electrospun, poly(acrylic acid-co-acryloylmorpholine), nanofiber, hydrophilic, antibacterial, wound healing, BSA-affinity

## Abstract

In this study, poly(AA-co-ACMO) and polyurethane-based nanofibers were prepared in a ratio of 1:1 (NF11) and 2:1 (NF21) as antimicrobial carriers for chronic wound management. Different techniques were used to characterize the nanofibers, and poly(AA-co-ACMO) was mostly found on the surface of PU. With an increase in poly(AA-co-ACMO) dose from 0 (PU) and 1:1 (NF11) to 2:1 (NF21) in the casting solution, the contact angle (CA) was reduced from 137 and 95 to 24, respectively, and hydrophilicity was significantly increased. As most medications inhibit biological processes by binding to a specific protein, in vitro protein binding was investigated mechanistically using a stopped-flow technique. Both NF11 and NF21 bind to BSA via two reversible steps: a fast second-order binding followed by a slow first-order one. The overall parameters for NF11 (*K*_a_ = 1.1 × 10^4^ M^−1^, *K*_d_ = 89.0 × 10^−6^, Δ*G*^0^ = −23.1 kJ mol^−1^) and NF21 (*K*_a_ = 189.0 × 10^4^ M^−1^, *K*_d_ = 5.3 × 10^−6^ M, Δ*G*^0^ = −27.5 kJ mol^−1^) were determined and showed that the affinity for BSA is approximately (NF11)/(NF21) = 1/180. This indicates that NF21 has much higher BSA affinity than NF11, although BSA interacts with NF11 much faster. NF21 with higher hydrophilicity showed effective antibacterial properties compared to NF11, in agreement with kinetic data. The study provided an approach to manage chronic wounds and treating protein-containing wastewater.

## 1. Introduction

Chronic skin wounds impose a significant medical burden on patients, particularly those with burns [1,2,3,4]. A good skin wound dressing must satisfy the following criteria [5]: good tissue compatibility, which avoids toxicity or inflammation [6,7], and good moisture retention, which keeps the wound moist and increases cell hydration [8], as well as suitable physical and mechanical durability, which ensures the dressing’s integrity and prevents external bacterial infection caused by material damage [9].

For many years, N-acryloylmorpholine (ACMO) and its high molecular weight polymers have been researched [10]. ACMO is widely used in peptide synthesis, enzyme immobilization, blood plasma separation nanofibers, and drug release applications. Because ACMO has a disubstituted amide group, it can act as an acceptor in a hydrogen bond [11]. ACMO is a non-ionic, hydrophilic monomer and is also perfect for increasing the swelling ratio of hydrogels, allowing water to penetrate the polymeric network [12]. Poly(*N*-acryloylmorpholine) (PACMO) is a hydrophilic polymer with unique features such as minimal toxicity and inhibition of in vivo immunological reactions [13,14]. Because of its high biocompatibility with the human body, PACMO has been employed in the transdermal delivery of active drugs [15]. Previous research has also shown that the PACMO-functionalized solid surface can reject non-specific protein adsorption [16,17,18]. PACMO’s poor affinity for protein can be attributed to properties such as hydrophilicity, hydrogen bond acceptors, a lack of hydrogen bond donators, and electrical neutrality [19]. Electrospinning has evolved into a versatile technology for producing a wide range of fibers with diverse desired functions from a wide range of materials, including polymer–polymer blends [20,21,22,23,24], sol–gels [25,26], composites [27], and ceramics [28]. Owing to its good hydrophilicity and its polymer backbone with ionic pendant groups, acrylic acid (AA) was used by Sinah et al. to obtain a novel pH-responsive hydrophilic cross-linked poly(acrylic acid-co-acryloylmorpholine) (poly(AA-co-ACMO)) copolymer [29]. Because of its excellent chemical stability, adjustable mechanical performance, and outstanding biodegradability, polyurethane (PU) material has received much attention [30,31,32,33]. Grafting PU and poly(AA-co-ACMO) could be a way to combine the mechanical properties of PU with the hydrogel properties of poly(AA-co-ACMO). The addition of hydrophobic PU can also improve the longevity of poly(AA-co-ACMO) nanofibers in water [34].

Proteins are the most important biomolecules, and they are involved in a wide range of biological processes [35]. Protein–drug interactions, as well as protein structure serum albumin, which is found in blood plasma, are essential for drug transport and disposition in the biological system. Bovine serum albumin (BSA) has been extensively studied as a model protein for these interactions due to its structural similarity to human serum albumin [36]. Because its structure is similar to that of human serum albumin, the chosen BSA is an excellent model protein for studying interactions. 

In this study, poly(AA-co-ACMO) in different ratios was blended with PU, and poly(AA-co-ACMO)/PU nanofibers (NF11 and NF21) were obtained using the electrospinning technique. A scanning electron microscope and FTIR spectroscopy were used to examine the surface morphology of the nanofibers and functionalization success, respectively. Furthermore, the thermal, contact angle measurements, and mechanical properties of the nanofibers were investigated. BSA-binding properties were studied to evaluate their pharmacological actions. Because protein–drug interaction is thought to be a fast step, we used a stopped-flow technique to analyze the binding process [37,38,39,40]. This technique aids in determining the kinetic characteristics of stages leading to intermediate species and, ultimately, obtaining an interaction mechanism. Additionally, the antibacterial activity is also reported and correlated with the kinetic parameters to evaluate the properties of electrospun nanofibers for chronic wound-dressing applications.

## 2. Results and Discussion

### 2.1. Characterization of Poly(AA-co-ACMO)/PU Nanofibers 

Electrospun poly(AA-co-ACMO)/PU nanofibers were made in a 1:1 (*w*/*w*) (NF11) and 2:1 (*w*/*w*) (NF21) ratio, respectively, using electrospinning technology (See Figure 1).

The nanofibers PU, NF11, and NF21 were synthesized and characterized by FTIR in order to confirm the existence of functional groups in the copolymer, and Figure 2 shows the FTIR spectra of NF11 and NF21 nanofibers, as well as PU and poly(AA-co-ACMO)copolymers. The stretching vibrations of the -C-N bond present in PACMO are associated with the characteristic peak at 1509 cm^−1^ [41]. Furthermore, the peaks at 1256 cm^−1^ and 1098 cm^−1^ are caused by the stretching vibration of the C-O-C bond in the ACMO’s morpholine ring. The stretching vibration of C=O of ACMO is responsible for another peak at 1660 cm^−1^ (NF11) and 1666 cm^−1^ (NF21). The spectra also show a peak around 1723 cm^−1^ that is attributed to the -COOH group in AA. The stretching vibration of the acrylic carboxylic –OH group was attributed to the peak at 3479 cm^−1^ observed in the region 3000–4000 cm^−1^. Other peaks at 2932 and 2877 cm^−1^ are assigned for asymmetric stretching of the methylene groups (C-H of CH_2_). A hydrogen-bonded carbonyl urethane group [42] was assigned to the presence of the peak at 1700 cm^−1^. As a result, the FTIR spectra revealed that hydrogen bonding took place in these nanofibers. Thus, the FTIR data of both synthesized NF11 and NF21 fibers contain all PACMO, AA, and PU characteristic peaks that are slightly shifted, indicating the presence of ACMO, AA, and PU in the nanofibers. Due to the increased feed ratio of poly(AA-co-ACMO), the intensity of all characteristic peaks of ACMO and AA was high in the NF21 nanofiber.

### 2.2. Hydrophilicity Analysis 

Static water CA was used to determine the nanofibers’ hydrophilicity. The hydrophilicity (lower hydrophobicity) of nanofibers increases as the CA value decreases and vice versa. Table 1 shows the static water contact angle of PU, NF11, and NF21 nanofibers measured with a goniometer, as well as the applied water droplets [25]. With an increase in poly(AA-co-ACMO) dose from 0, 1:1 to 2:1 in the casting solution, the CA is reduced from 137, 95 to 24° for PU, NF 11, and NF 21, respectively. The trend of the water CA of nanofibers was PU > NF11 > NF21, indicating that the CA decreased with the increase in poly(AA-co-ACMO) dose. These results show that the hydrophilicity of the NF21 nanofibers increased significantly by the addition of poly(AA-coACMO) content. The increased hydrophilicity of the nanofibers was entirely due to the poly(AA-co-ACMO)’s inherent hydrophilicity. The presence of the terminal carboxyl groups of the acrylic acid on the fibers considerably enhanced the hydrophilicity of the poly(AA-coACMO)/PU nanofibers’ surface. Hydrophilicity increases the penetration of the liquid into the porosity to storage the water and as a consequence increases the uptake ability. By virtue of the CA results, the degree of hydrophilicity of the NF 21 nanofibers in the current study was relatively higher than the hydrophilicity found in previous studies [29,43].

### 2.3. Morphology of Poly(AA-co-ACMO)/PU Blend Nanofibers 

Electrospun nanofibers were used to assess the effect of poly(AA-co-ACMO on fiber morphology, and scanning electron microscopy (SEM) was used to investigate the difference in morphology between the electrospun nanofibers (Figure 3). Figure 3 displays SEM images of PU, NF11, and NF21 nanofibers, and more pictures with different magnifications are also presented in the Appendix A. The structure and diameter of the electrospun nanofibers were greatly influenced by the variation in poly(AA-co-ACMO) concentrations. Micrographs of fibers electrospun at various poly(AA-co-ACMO) concentrations are shown in Figure 3. 

The electrospinning process was observed to produce fibers in the absence of poly(AA-co-ACMO) (pure PU); however, as the concentration of poly(AA-co-ACMO) increased, electrospinning became more difficult due to the higher viscosity of the resulting solution, and the electrospun nanofiber displayed further non-smoothness and demonstrated a gluing between nanofibers (Figure 3b,c). The agglomeration of poly(AA-co-ACMO) and subsequent creation of the gluing nanofibers were brought on by the high viscosity and excess poly(AA-co-ACMO). The results suggested that the morphology of the electrospun nanofibers was influenced by the electrospinning solution’s immiscibility. The SEM image of the PU is shown in Figure 3. As can be seen from this figure, the resulting fibers are quite uniform and smooth with a few structural defects. The increasing addition of poly(AA-co-ACMO) to PU leads to a decrease in the number of fine fibers, and some kinds of core–shell-like structures are formed, as shown in Figure 3a,b [44]. The findings suggested that PU, rather than poly(AA-co-ACMO), was the nanofiber’s backbone [45].

### 2.4. Mechanical Properties (Tensile Strength) Measurement

The mechanical properties of materials are known to be strongly influenced by microstructure and polymer interaction. The study of this property would yield valuable information about the internal structure of materials [46,47]. Figure 4 depicts the stress–strain behavior, Table 1 summarizes the mechanical properties, and more details are also presented in the Appendix A. Tensile measurements were taken in both the dry and wet states to assess the mechanical properties of the typical stress–strain curves of PU, NF11, and NF21 nanofibers. 

The results reveal that improving the poly(AA-co-ACMO) content of the PU matrix enhances the mechanical properties. The best nanofiber is NF11 (loaded at a ratio of 1:1), which has a breaking point strain of 9.16% and an ultimate strength of 15.83 MPa. The nanofiber NF11 is therefore thought to have a stronger ability to resist compression deformation. It is important to note that adding more poly(AA-co-ACMO) does not raise or diminish the nanofibers’ mechanical characteristics. Therefore, the poly(AA-co-ACMO) contents in the nanofiber should be adjusted in order to produce nanofibers with high mechanical performances. 

It is understood that a significant interaction between the PU and poly(AA-co-ACMO) molecules is facilitated by the presence of many carboxyl groups on the poly(AA-co-ACMO) surface. Therefore, homogenous dispersion of the poly(AA-co-ACMO) in the PU matrix is preferred. When the amount of poly(AA-co-ACMO) is low, the interaction between the poly(AA-co-ACMO) molecules is weaker than the interaction between the poly(AA-co-ACMO) and the PU molecules, allowing the poly(AA-co-ACMO) to be dispersed uniformly throughout the PU matrix. The distance between poly(AA-co-ACMO) is shortened as the poly(AA-co-ACMO) loadings rise and the contact intensifies. Because of the aggregation, the poly(AA-co-ACMO) reinforcing function is lessened [48]. As a result, nanofibers’ mechanical characteristics are enhanced. A strong interfacial adhesion and poly(AA-co-ACMO) dispersal and interaction with the polymer matrix are also essential for the poly(AA-co-ACMO) to make important contributions. On the other hand, the cross-linking of polymers may be prevented by the addition of a significant amount of poly(AA-co-ACMO). As a result, the cross-linking density decreases, increasing the quantity of equilibrium water that may sorb. Second, the polymerization of the monomers is hindered by the polymerization of the poly(AA-co-ACMO), which results in an inhomogeneous structure for the nanofibers.

According to the aforementioned description, low-poly(AA-co-ACMO) loadings (NF11) enhance the nanofibers’ mechanical qualities, while high-poly(AA-co-ACMO) loadings cause them to decline (NF21). As a result, the concentration of poly(AA-co-ACMO) should be kept within reasonable limits [49,50].

### 2.5. Thermal Stability of the Composite Nanofibers

TGA was used to investigate PU, NF11, and NF21, as shown in Figure 5, which depicts the thermograms of nanofibers with two stages of thermal degradation. Until about 250 °C, all curves demonstrated stability and heat resistance. As the temperature rose further, the nanofibers began to degrade. The thermograms of NF11 and NF21 nanofibers were nearly identical, with a first stage at around 305 °C attributed to acrylic acid degradation. The ACMO moiety is degraded in the second stage, which occurs at 411 °C. PU11 had the highest *T_g_* and the best TGA performance. A slightly higher mass retention for PU was observed across the entire temperature range, particularly between 250 and 600 °C, indicating better heat resistance in the lower temperature range. The first and second stages of PU were demonstrated at slightly higher temperatures (316, 416 °C) than both NF11 and NF21, indicating that PU has greater thermal stability. DTG thermograms (Figure 5) clearly showed that at around 300 °C, degradation of PU the copolymer was suppressed. As dw/dt was zero, the TG curves reached a plateau around 530 °C. Practically complete degradation is the fastest for PU and the char yield reached the lower value equal to 3%, whereas NF21 is the slowest and the char yield reached the higher value equal to 5%. 

### 2.6. Antibacterial Activity Study

PU, NF11, and NF21 nanofibers were tested for antibacterial activity against *Staphylococcus aureus* and *Enterococcus faecalis* using the viable cell count method [51]. In this section, the minimal bactericidal concentration (MBC) is the lowest concentration of tested biocidal agents or fillers used to fill nanofiber mats that kill 99.9% of the initial bacterial colony. The relationship between the mean inhibition zone diameter and nanofiber concentrations is shown in Figure 6. The hydrophilic NF21 composite exhibited more effective antibacterial activity than the hydrophobic PU and NF11 composites, according to the antibacterial activity results. The formation of a core–shell-like fiber structure in the NF21 composite improved the hydrophilicity of PU. The nanofibers’ adsorption onto the bacterial cell, diffusion through the cell wall, binding to the cytoplasmic membrane, release of cytoplasmic constituents such as K ions, DNA, and RNA, and cell wall damage disrupt the cell’s natural processes and ultimately result in the rapid death of microorganisms in these dual actions.

### 2.7. Mechanistic Investigation of BSA Binding 

All of the studies have paid close attention to compounds that are linked to BSA, which appears to be a necessary but not the only factor influencing biological activity. Affinity, kinetics (rates) of association, and dissociation are all critical for biological activity [52]. These studies used the slowed-flow technique. The kinetic traces for the interaction with BSA can be observed at 270 nm. The kinetic curves cover the entire process and can be fitted to a two-exponential function, implying that at least two binding steps are involved, as shown by the UV–Vis spectra in Figure 7 as well as the kinetic traces shown in Figure 8. The first interaction step was found to be significantly faster than the second; 10^−4^ mol/L solutions of BSA were reacted with various concentrations of compounds while maintaining the pseudo-first-order reaction rate to see if the observed reaction rates were concentration-dependent. The plot of *k*_obs_ versus the concentration of NF11 and NF21 yielded a linear plot with a slope equal to *k*_1_ and an intercept equal to *k*_-1_, and the observed rate constant can be expressed by Equation (2) (Figure 9).
*k*_obs_ = *k*_1_ + *k*_−1_ [Nanofibers](1)

Here, *k*_1_ and *k*_−1_ are rate constants that describe the first step kinetics of BSA complex formation as well as dissociation. The first steps in Figure 1 express this fast and reversible step of BSA binding, which includes compound formation and dissociation.

BSA binds reversibly to NF11 with a second-order association constant of *k*_1_ = 45.7 ± 4.5 M^−1^s^−1^ in the initial phase and dissociates from the binary complex with a first-order dissociation constant of *k*_−1_ = 0.257 ± 0.013 s^−1^. In contrast, NF21 binds to BSA eight times slower (*k*_1_ = 6.2 ± 0.4 M^−1^s^−1^) and dissociates about thirty times slower (*k*_−1_ = 7.3 ± 1.3 s^−1^). This means that BSA’s binding affinity for NF11, *K*_a1_, *k*_1_/*k*_−1_ = 178 M^−1^ is about five times lower than that of NF21 (827 M^−1^) and that BPA’s equilibrium dissociation constant, *K*_d1_, *k*_-1_/*k*_1_ = 6 × 10^−3^ M, is about five times higher than that of NF21 (1 × 10^−3^ M). This means that the NF21–BSA that has been formed is far more stable than NF11–BSA. The equilibrium dissociation constant, *K*_d1_, is related to the difference between the free energy, *G*, of BSA molecules alone in the solution and when bound together. The binding free energy change *G*_bind_ is calculated using Equation (3).
(2)$$\Delta G_{bind}=RT\cdot ln(K_{d})$$

While *T* and *R* represent the absolute temperature and the universal gas coefficient, respectively, the Δ*G_bind_* values, calculated using Equation (4), are −12.6 and −17.1 kJ mol^−1^ for the first step binding of BSA to NF11 and NF21, respectively. The negative sign for *G* indicates spontaneous binding to both NF11 and NF21, and the greater negative value of NF21 compared to NF11 indicates that binding with NF21 is very favorable. The data from the second reaction step, primarily concerned with the BSA condensation process, in addition to the electrostatic interaction, followed the same pattern as the first; a reversible reaction was observed for both NF11 and NF21. The BSA affinity for NF11 (*K*_a2_ = *k*_2_/*k*_−2_) is 65.9 M^−1^, which is about thirty times lower than NF21 (1900 M^−1^), and the equilibrium dissociation constant for NF11 (*K*_d2_ = *k*_−2_/*k*_2_) is 15.0 × 10^−3^ M, which is about thirty times faster than NF21 (0.53 × 10^−3^ M). G*_bind_* values for this phase are −6.7 kJ mol^−1^ (NF11) and −18.7 kJ mol^−1^ (NF21), indicating that binding with NF21 is very favorable as the first step. Equation (4) shows how to calculate the overall equilibrium dissociation constant, *K_d_*, from the individual equilibrium dissociation constants, *K_d_*_1_ and *K_d_*_2_, as well as the overall association constants, *K_a_*.
(3)$$K_{d}=\frac{K_{d1}K_{d2}}{1+K_{d2}}$$

The data in Table 2 show that the overall BSA binding affinity of NF21 (189 × 10^4^ M) is much higher than that of NF11 (1.1 × 10^4^ M). The overall reaction *G_bind_* values are −23.1 kJ mol^−1^ (NF11) and −27.5 (NF21) kJ mol^−1^, indicating that the reaction is spontaneous for both compounds. This means that when the dose of copolymer poly(AA-co-ACMO) in the casting solution was increased, the value of BSA binding affinity increased. These findings show that blending poly(AA-co-ACMO) copolymers enhanced the BSA affinity at the nanofibers’ interface. Increases in hydrophilicity of poly(AA-co-ACMO)/PU (decrease in CA) were the cause, as previously stated. The poly(AA-co-ACMO) content was higher in NF21 nanofibers and lower in NF11 nanofibers, resulting in a BSA affinity trend of NF11 > NF21. The following factors could explain this phenomenon: van der Waals interactions between water and the membrane surface; electrostatic repulsion between the membrane surface and protein [50] due to the hydrophilic layer; and hydration interactions as a result of water layer formation linked to the hydrophilic additive that produces a repulsive force towards protein [29,43].

## 3. Materials and Methods 

### 3.1. Materials 

The following items were purchased: ACMO (Aldrich, Milwaukee, WI, USA), AA (Merck, Stuttgart, Germany), 2,2-azoisobutyronitrile (AIBN) as an initiator, and PU (Noveon BVBA, Chaussee de Wavre 1945, B- 1160 1160 Brussels). The rest of the reagents were used without purification. A BSA stock solution (1 × 10^−3^ mol/L) was prepared in Tris buffer at pH 7.4 using bovine serum albumin (BSA, Sigma-Aldrich, Burlington, MA, USA). Solutions (4 × 10^−4^ mol/L) were prepared in Tris buffer from the first stock solution to a lower concentration of ethanol during sequential addition in titrations with BSA (to avoid ethanol’s effect on BSA structure, concentrations of ethanol were kept below 1% (wt/wt) for all titrations [53]).

### 3.2. Synthesis Poly(AA-co-ACMO) Copolymer

Poly(AA-co-ACMO) was synthesized as previously reported [54]. In a nutshell, acryloyl morpholine (ACMO) (5 mL, 0.04 mol), acrylic acid (AA) (3 mL, 0.04 mol), and azoisobutyronitrile (AIBN) (0.0651 g, 0.00039 mol) were dissolved in 10 mL of dry toluene. The polymerization mixture was kept at 70 °C for 5 h under N_2_ atmosphere. The polymer was then filtered, washed with methanol, and dried at 40 °C until a consistent weight was achieved. The success rate was 95%.

### 3.3. Fabrication of poly(AA-co-ACMO)/PU (NF11 and NF21) 

To form uniform solutions, poly(AA-co-ACMO) and PU were dissolved in a mixture of chloroform/dichloromethane solution (7 mL:3 mL) by stirring for 8 h at room temperature. Poly(AA-co-ACMO) to PU component ratios were 1:1 (*w*/*w*) and 2:1 (*w*/*w*). The solution was dispensed into a 10 mL syringe fitted with a 0.25 mm inner diameter metal needle. The solution flow rate was set to 4 mL h^−1^ with a 60 kV applied voltage. The tip-to-collector distance was 16 cm, and the collector was a grounded aluminum sheet. The electrospun nanofiber film was dried in an air oven at 50 °C for 24 h to thoroughly eliminate the residual solvent.

### 3.4. Morphology and Characterization of Nanofibers

A field emission electron microscope (SEM) was used to examine the surfaces of the electrospun nanofibers (TESCAN Vega 3, 5 kV, 5 mm working distance, SE detector). The nanofibers were identified using FTIR spectra recorded at room temperature with a spectrometer (Perkin–Elmer, Waltham, MA) and measured in the wavenumber range 4000 to 400 cm^−1^ by dispersing them in KBr matrices. The thermal behavior of the nanofiber was studied using thermogravimetric analysis (TGA, STA449F3; NETZSCH, Selb, Germany). Under a nitrogen atmosphere, the heating rate is 10 °C/min, with a heating range of 25 to 800 °C.

### 3.5. Contact Angle Measurement

Under ambient conditions, the dynamic contact angles of nanofibers were measured under ambient conditions using a contact angle meter (DSA 25; KRUSS, Hamburg, Germany). Deionized water was chosen as the liquid for contact angle measurement. A water droplet (0.05 mL) was placed on the nanofiber membrane’s surface, and the contact angle was measured for 1 to 30 s.

### 3.6. Mechanical Measurement

Mechanical tensile tests were performed with a TA Instruments Discovery Hybrid Rheometer D-HR3 equipped with linear tensile geometries dedicated to film characterization. Large scaffolds were cut into rectangular samples with a 5 mm width, 22 mm length, and a thickness of less than 100 microns and clamped. Each sample’s thickness was meticulously measured using a scanning electron microscope (SEM) examination of each cross section. The sample length between the clamps was initially set to 10 mm, the tensile velocity was set to 50 m/s, and the temperature was set to 21 °C. The samples were first fixed in the clamps at dry state and then hydrated in distilled water before starting the tensile test to characterize the scaffolds in the wet state.

### 3.7. Antibacterial Ability Test

The viable cell count method [55] was used to assess the antimicrobial ability of PU, NF11, and NF21 nanofibers. The antibacterial activity of the prepared nanofibers was tested using two bacterial strains (*E. faecalis* and *E. coli*). Bacterial strains were inoculated in LB broth medium containing Tryptone (1%), yeast extract (0.5%), and NaCl (1%) at pH 7 and incubated overnight at 37 °C in a shaking incubator (Model InnovaTMJ-25, New Brunswick Scientific, Edison, NJ, USA) for refreshment. With LB broth medium (1%), the bacteria suspensions were diluted up to 100 times. A volume of 100 L of diluted suspension was cultured in 10 mL of Tryptone medium containing 0.1 g of tested sample and then sterilized for at least 20 min at 121 °C. The mixture was then left to sit for a while. After that, the mixture was shaken for 18 h at 37 °C, and the percentage of bacteria growth inhibition was determined by measuring the absorbance of the culture medium at 600 nm using visible spectroscopy. The inhibition (percentage) was calculated using the equation below:Inhibition (%) = (*A*_a_ − *A*_b_)/*A*_a_ × 100(4)
where *A*_b_ and *A*_a_ are the absorbance of bacterial culture in absence and in presence of tested sample, respectively.

### 3.8. BSA Binding Studies

In methanol, stock solutions of nanofibers (10^−3^ mol/L) were prepared, which were then diluted to make working solutions for spectroscopic methods. Phosphate-buffered solution (PBS) at pH 7.4 was used to make all of the solutions. All of the experiments were carried out at a pH of 7.4 and at room temperature. At 240–360 nm, the absorption spectra of BSA were recorded in the absence and presence of increasing amounts of nanofibers. The spectral changes in the reactions were first recorded over the wavelength range of 200–700 nm for stopped-flow measurements, and the kinetic traces were recorded at 270 nm. Using at least a 10-fold number of nanofibers, all reactions were studied under pseudo-first-order conditions. By fitting absorbance time traces to a two-exponential function, the pseudo-first-order rate constant was investigated. Under each set of experimental conditions, all listed rate constants are the average of at least two independent kinetic runs.

## 4. Conclusions

Poly(AA-co-ACMO) and PU-based electrospun nanofibers are being developed for use as antimicrobial carriers in chronic wound treatment. To improve the hydrophilicity of electrospun PU nanofibers, various amounts of poly(AA-co-ACMO) were added to PU, and poly(AA-co-ACMO) was mostly found on the PU nanofibers’ surfaces. Increasing the dose of poly(AA-co-ACMO) in the casting solution reduces contact angles, thereby enhancing the hydrophilicity of the surface of poly(AA-co-ACMO)/PU nanofibers. Because almost all pharmaceuticals work by interfering with biological functions by binding to a specific protein, in vitro protein binding of both nanofibers was studied mechanistically using a stopped-flow technique and a mechanism was proposed. NF11 and NF21 bind to BSA in a similar way, with two reversible steps: a fast second-order binding followed by a slow first-order one. The overall binding parameters were determined and showed that the relative affinity for BSA is approximately (NF11)/(NF21) = 1/180. This suggests that, despite the fact that BSA interacts with NF11 significantly faster, NF21 has a much higher affinity for BSA. This is because the rapid association of NF11 with BSA is followed by a fast dissociation reaction. Higher *K*a values (10^5^ to 10^7^ M^−1^) compared to that reported in the literature are associated with high degrees of binding (90% to 99.9%) [56,57]. The increased hydrophilicity nanofibers NF21 demonstrated good antibacterial capabilities compared to the less hydrophilic NF11, which is consistent with the protein-binding affinity. These results can be used to estimate the impact of protein binding on antimicrobial killing for various antibiotics and bacteria [58], and this research could lead to a new method of managing chronic wounds and treating protein-containing wastewater.

## Data Availability

Not applicable.

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
