# Peer review of "Microstructure, Physical and Biological Properties, and BSA Binding Investigation of Electrospun Nanofibers Made of Poly(AA-co-ACMO) Copolymer and Polyurethane"

_molecules, 2023, doi:10.3390/molecules28093951_

Round 1
Reviewer 1 Report
In this paper, entitled “Microstructure, physical and biological properties, and BSA binding investigation of electrospun nanofibers made of poly(AA-co-ACMO) copolymer and polyurethane”, Eldik et al. synthesized electrospun nanofibers based on poly(AA-co-ACMO) and PU at different ratios of NF11 and NF21. The nanofibers were designed for use as antimicrobial carriers in chronic wound treatment. By incorporating varying amounts of poly(AA-co-ACMO), the authors were able to improve the hydrophilicity of the PU electrospun nanofibers. In addition, the authors proposed a mechanism for the in-vitro protein binding of both types of nanofibers. The results showed that the composite nanofibers NF21, which had higher hydrophilicity, exhibited increased antibacterial properties compared to the less hydrophilic NF11. This finding is consistent with the stopped-flow kinetic data. Overall, the content of the paper is interesting and suitable for publication. However, there are some minor points of concern that need to be addressed prior to acceptance.
1. The authors conducted a comprehensive investigation of electrospun nanofibers made of poly(AA-co-ACMO)/PU at both a 1:1 and 1:2 ratio. It is unclear from the paper whether they examined other ratios. However, it is worth noting that the antibacterial activity of the nanofibers is primarily influenced by the copolymer concentration. Therefore, it would be useful to compare the impact of various copolymer concentrations on the performance of the nanofibers.
2. The image presented in Figure 6 is difficult to discern. To improve clarity, I suggest increasing the resolution and reformatting the image.
3. The proposed antibacterial mechanism of these nanofibers, as outlined on page 14, requires revision. While the authors note that most pharmaceuticals work by binding to specific proteins to interfere with biological functions, it would be beneficial to compare this paper with other relevant studies to highlight the advantages of this approach. This would help to clarify the unique contribution of the current work in the field of antibacterial materials.
4. It would be beneficial to cite some related works on the application of polymers, such as, Small 2022, 18 (16), 2106893.; J. Catal. 2021, 396, 342.
5. The formatting of the Figures in the manuscript needs to be double-checked for errors. For instance, it is important to perform a case-sensitive check for Figure 4, and ensure that the labels (a) and (b) are included for all sub-figures in Figure 9. Correcting these issues will improve the clarity and accuracy of the manuscript.
No.
Reviewer 2 Report
The manuscript entitled “Microstructure, physical and biological properties, and BSA binding investigation of electrospun nanofibers made of poly(AA-co-ACMO) copolymer and polyurethane” submitted by Mansour et al. reports the obtention of nanofibrous mats based on the combination between P(AA-co-ACMO) and PU for the desired application in the field of antimicrobial wound healing materials. The authors synthesized the aforementioned polymer and then obtained the electrospun mat studying the morphology, wettability, mechanical properties, antibacterial activity, and BSA binding. Even though I consider the research interesting to be published in Molecules journal, I believe that the authors should improve some key aspects of the manuscript prior to acceptance. I will give some suggestions and important comments below.
1- Abstract. Define CA. I also believe that “tools” should be replaced by “techniques”.
2- Introduction. “In vivo” should be in italics.
3- Introduction. Last Paragraph: FT-IR is used to determine functionalization success but not for surface study. Please, modify adequately.
4- The authors should homogenize units throughout the whole manuscript. E.g., mol/L or mol/l, ml or mL, microns, l, percent, etc.
5- Section 2.1. Ethanol concentrations were kept below 1%... in mass? Volume?
6- Section 2.2. Please, use scientific notation when necessary. Also, clarify N2 (g). Regarding polymer filtration, some specifications are missing.
7- Section 2.3. Why did the authors say “hydrogel”? Is it a crosslinked network?
8- Section 2.3. Please, write the ratio between solvents adequately.
9- Section 2.3. Please, inform me about environmental and solution parameters. By the way, I suggest the authors include in supporting information the measurements related to the viscosity and conductivity of the sample’s prior electrospinning process.
10- Section 2.4. The samples were examined after gold sputtering.
11- Section 2.5. Please, indicate the volume of water drops.
12- Section 2.7. Please, write “1 percent” adequately.
13- Figure 1. Replace "5 hs" by "5 h".
14- Figure 2. For better comprehension, indicate in the graph all the important peaks and avoid superposition between the different spectra.
15- Section 3.2. I strongly suggest that the authors incorporate the pictures of water drops on the surface in the supporting information of the manuscript.
16- Section 3.2. When the authors inform the diameter of nanofibers. I would like to know how the determination was made. How many fibers did the authors count? Please, add the standard deviation and, if possible, add the histogram of distribution.
17- Section 3.2. The authors referred to viscosity solution influence. I strongly suggest the inclusion of the data.
18- Figure 3. SEM images should be improved in quality, especially b, and c. It would be great if the authors showed different magnifications.
19- Table 1. Use correct significative numbers. Consider also comment number 16.
20- TGA results. The residual mass should be informed and discussed.
21- How do the authors ensure the formation of a core-shell fiber structure in section 3.6?
22- Figure 6. Please, add error bars, and the units and use an italic letter for bacteria strain.
23- Page 11. Homogenize “M” or “mol/L”.
24- Section 3.7. The authors referred to Equation 6 but is not appear.
25- Table 4. Homogenization of units is required.
26- Conclusion. Clarify the fact that it is a POTENTIAL application in chronic wound treatment. Also, I suggest modification and improvement of this section due to seems so like the abstract.
27- Conclusion. The authors referred to a composite… is it a true composite? Justify.
-
Round 2
Reviewer 1 Report
All of my concerns have been addressed by the authors. Therefore, I believe that this paper is ready for publication.
Reviewer 2 Report
The authors solved the asked issues. I have not more comments.